# Maternal Phenylketonuria and Offspring Outcome: A Retrospective Study with a Systematic Review of the Literature

**DOI:** 10.3390/nu17040678

**Published:** 2025-02-14

**Authors:** Guido Leone, Concetta Meli, Raffaele Falsaperla, Federica Gullo, Laura Licciardello, Luisa La Spina, Marianna Messina, Manuela Lo Bianco, Annamaria Sapuppo, Maria Grazia Pappalardo, Riccardo Iacobacci, Alessia Arena, Michele Vecchio, Martino Ruggieri, Agata Polizzi, Andrea Domenico Praticò

**Affiliations:** 1Postgraduate Training Program in Pediatrics, University of Catania, 95125 Catania, Italy; 2Unit of Expanded Neonatal Screening and Inherited Metabolic Diseases, Pediatric Clinic, Department of Medical Sciences, University of Catania, 95123 Catania, Italy; 3Unit of Pediatrics, University of Ferrara, 44121 Ferrara, Italy; 4Unit of Pediatric Clinic, Department of Clinical and Experimental Medicine, University of Catania, 95123 Catania, Italy; 5Unit of Pediatrics and Pediatric Emergency, AOU “Policlinico” PO “San Marco”, 95123 Catania, Italy; 6Rehabilitation Unit, Department of Biomedical and Biotechnological Sciences, University of Catania, 95123 Catania, Italy; 7Unit of Pediatrics, Department of Medicine and Surgery, University Kore of Enna, 94100 Enna, Italy

**Keywords:** phenylketonuria, diet, maternal therapy, phenylalanine, outcome

## Abstract

**Background:** Phenylketonuria (PKU) poses significant challenges for maternal and neonatal outcomes, requiring strict adherence to dietary protocols to maintain optimal maternal phenylalanine (Phe) levels during pregnancy. This study retrospectively analyzed outcomes of pregnancies in PKU-affected women and conducted a systematic review on the timing of dietary management and its impact on outcomes. **Methods:** This retrospective study included data from nine PKU-affected women and 14 pregnancies followed at the Regional Reference Center for Metabolic Diseases in Catania. Women were categorized based on the timing of dietary intervention: preconception (pcD), during pregnancy (pD), or never (nD). Maternal Phe levels were classified as in-target (tP+) or non-target (tP−). A systematic review of the literature was conducted using PRISMA methodology, including 77 studies reporting maternal Phe levels, dietary timing, and clinical/auxological offspring outcomes. **Results:** In the retrospective study, pcD and tP+ groups had significantly better neonatal outcomes, with lower rates of congenital heart disease (CHD), facial dysmorphisms, intrauterine growth restriction (IUGR), and microcephaly. Systematic review data from 1068 PKU-affected women and 2094 pregnancies revealed that pcD with tP+ resulted in the lowest rates of miscarriage (0.14%) and adverse neonatal outcomes, while tP− and nD groups showed the highest rates of CHD, microcephaly, and intellectual disability. **Conclusions:** Early dietary intervention, ideally preconceptionally, and achieving target maternal Phe levels are critical in reducing adverse outcomes in pregnancies of PKU-affected women. These findings emphasize the importance of metabolic control and adherence to dietary protocols in maternal PKU management.

## 1. Introduction

Phenylketonuria (PKU) is a rare inborn error in phenylalanine (Phe) metabolism mostly caused by autosomal recessive mutations in the phenylalanine hydroxylase (PAH) gene. Tetrahydrobiopterin (BH4) is the necessary cofactor for PAH conversion of Phe into tyrosine (Tyr); thus, variants in the genes encoding the enzymes involved in BH4 metabolism also result in increased hematic levels of Phe, known as hyperphenylalaninemia (HPA) [1,2,3].

PKU became the first condition to benefit from newborn screening (NBS). Nevertheless, the NBS is for HPA, which is indicated by any blood Phe level greater than 120 μmol/L [3]. Four different phenotypes of PAH deficiency have been defined: “classic PKU”, which presents with Phe pretreatment levels above 1200 μmol/L; “moderate PKU”, which presents with Phe pretreatment levels of 900–1200 μmol/L; “mild PKU”, presenting with Phe pretreatment levels of 600–900 μmol/L; and “mild HPA”, which presents with Phe pretreatment level less than 600 μmol/L [4,5,6,7].

Phe accumulates in the blood and central nervous system as a result of PAH deficiency. Without treatment, PKU causes an irreversible intellectual disability, microcephaly, motor impairments, eczematous dermatitis, autism, seizures, developmental delay, and psychiatric disorders [1,5]. All current treatments try to decrease blood Phe concentrations since high blood Phe levels are substantially correlated with poor neurocognitive outcome. A low-Phe diet combined with Phe-free L-amino acid supplementation is the basis of PKU treatment. Casein glycomacropeptide (GMP) and large neutral amino acids (LNAA) are two different dietary supplements that some PKU centers employ. BH4 (given as sapropterin dihydrochloride) acts as a pharmacological chaperone and is effective in treating some patients. Moreover, in 2018, the Food and Drug Administration (FDA) approved pegvaliase (Palynziq^®^): this is the only enzyme replacement therapy (ERT) currently available for individuals aged 16 years or older with an inadequate control of blood Phe levels (Phe values above 600 μmol/L) despite the previous treatment with other therapeutic options [6].

The goal of PKU therapy is also to avert maternal PKU syndrome in fertile women. Elevated blood Phe levels during pregnancy have teratogenic effects on the developing fetus and can cause facial dysmorphisms, congenital heart defects (CHDs), microcephaly, intellectual disabilities, and growth retardation [1]. Importantly, the need for a strict low-Phe diet in pregnant women is independent of the PKU status of the fetus, as high maternal Phe levels are harmful regardless of whether the fetus carries PAH mutations. The chances of a positive outcome are comparable to those of the general population when blood Phe levels are maintained within a certain range from preconception onwards (target Phe concentrations 120–360 μmol/L) [7]. This paper is intended to evaluate the clinical and auxological outcome of the offspring of PKU-affected women followed up in our Regional Reference Center of Metabolic Diseases in Catania, and not taking drugs, such as sapropterin or pegvaliase. Adjunctively, we performed a systematic review of the literature.

## 2. Material and Methods

### 2.1. Retrospective Study Design and Eligibility

This study contains a retrospective analysis about the clinical and auxological outcome of the offspring of PKU-affected women followed up in our Regional Reference Center of Metabolic Diseases in Catania. We primarily included the offspring of women meeting the following criteria:(I)PKU diagnosis before the onset of pregnancy.(II)At least one pregnancy carried out during lifetime, regardless of miscarriages or not.(III)No therapy or therapy consisting only of a low-Phe diet (i.e., without taking drugs, such as sapropterin or pegvaliase).

Relative to the clinical outcome of the offspring, we investigated for miscarriages, defined as spontaneous pregnancy losses in the first trimester or intrauterine fetal deaths (i.e., fetus losses after the third trimester); congenital heart diseases (CHD), defined as any affection of the heart present at birth; and facial dysmorphisms, i.e., facial anomalies of the nose, forehead, mandible, palate, or eyes.

Relative to the auxological outcome, we evaluated the weight and head circumference at birth. As regards the weight at birth, small for gestational age (SGA) was defined as birth weight < 2500 g at term or below the 10th percentile in preterm newborns; microcephaly was defined as a head circumference < 33 cm at term or below the 10th percentile in preterm newborns.

### 2.2. Data Collection of the Retrospective Study

Each woman underwent follow-up in our Regional Reference Center of Metabolic Diseases in Catania throughout her pregnancy. Diagnostic method of hyperPhe, age at diagnosis, and age at onset of pregnancy were primarily recorded. Then, medical history on pregnancy clinical course and gynecological consultations was collected prior to assess whether the woman followed the therapeutic dietary regimen and since when in relation to the start of pregnancy [i.e., no diet (nD); diet started from preconception (pcD); diet started in pregnancy (pD)]. Women taking drugs, such as sapropterin or pegvaliase, were excluded from this retrospective study. Blood Phe levels were tested at each follow-up. After childbirth, a massive clinical and auxological evaluation of the newborn was made. All data were retrospectively collected. The data collection time frame extends from 1 January 2000 to 1 March 2024.

### 2.3. Retrospective Study Patient Groups

According to various studies [1,7], dietary management should keep pregnant women in target concentrations between 120 and 360 μmol/L in order to avoid poor offspring outcome. Therefore, in step one, we divided our population of women into three groups, in consideration of whether they followed the dietary regimen or not (nD, pcD, or pD). Then, in step two, we integrated the blood Phe levels during pregnancy, identifying two groups: one with target Phe values (tP+), and one with non-target Phe values (tP−). Target Phe values were defined as mean values throughout pregnancy within 120–360 μmol/L. Finally, in step three, we assessed the clinical and auxological outcome of the offspring (dysmorphisms, CHD, microcephaly, intellectual disability, and growth retardation).

### 2.4. Search Strategy of the Systematic Review

Using the Preferred Reporting Items for Systematic Reviews and Meta-Analyses (PRISMA) standards [8], we also conducted a systematic review of the literature on the maternal PKU syndrome. We made three research attempts on PubMed/MEDLINE by applying the following filters together: case report, clinical trial, and randomized controlled trial. The three queries on PubMed were “Maternal phenylketonuria”, “Maternal PKU syndrome”, and “Phenylketonuria and pregnancy”, respectively. All the results refer to a timeline extending through Wednesday 24 January 2025.

### 2.5. Inclusion and Exclusion Criteria of the Systematic Review

In reference to the data collected in our retrospective analysis and according to the applied filters, any paper reporting (I) blood Phe levels during pregnancy, (II) information on the timing of starting the diet in relation to pregnancy, and (III) clinical and/or auxological details about the children of PKU-affected mothers met the inclusion criteria of our systematic review. Articles that did not address the aforementioned areas of interest, those giving data on PKU-affected women in therapeutic regimens other than diet, those only providing data on PKU-affected adults, non-English language studies, duplicates, views, reviews, systematic reviews, meta-analyses, statements, expert opinions, and guidelines were all excluded.

### 2.6. Study Selection of the Systematic Review

A five-step procedure based on removal of duplicates and non-English-language studies, screening via title and abstract, full-text reading, study selection according to inclusion/exclusion criteria, and data extraction was carried out. Elimination of duplicates was performed by one researcher; then, titles and abstracts were independently screened by two authors, who perfectly agreed on study elimination at the subsequent confrontation. Selected full-text articles were read by all the authors. Complete consensus was found regarding the papers to include in this study. Each researcher followed the inclusion and exclusion criteria unanimously decided in the pre-selection phase. Finally, all authors were responsible for data extraction.

### 2.7. Data Extraction of the Systematic Review

According to the data collected in our retrospective study, we extracted the following data from the included studies: total number of PKU-affected women, mean age at diagnosis, diagnostic method of hyperPhe (screening or not), total number of pregnancies, mean age at pregnancy onset, total number of born alive, total number of miscarriages, number of children with dysmorphisms, number of children with CHD, number of children with IUGR–SGA, number of children with microcephaly, and number of children with intellectual disability. As we did in the retrospective study, in the first phase, we separated the women into three groups based on whether or not they started the diet: diet started from preconception (pcD), diet started from pregnancy (pD), and no diet (nD). The blood Phe levels during pregnancy were then merged in step two, resulting in the identification of two groups: one with target Phe values (tP+) and one with non-target Phe values (tP−). In the third and final stage, we evaluated the offspring’s clinical and auxological outcomes (growth retardation, dysmorphisms, CHD, microcephaly, and intellectual disability). As said above, SGA was defined as a birth weight < 2500 g at term or below the 10th percentile in preterm newborns. Microcephaly was defined as a head circumference < 33 cm at term or below the 10th percentile in preterm newborns. CHD was defined as any affection of the heart present at birth. Intellectual disability was defined as an IQ below 70. Facial dysmorphisms refer to facial anomalies of the nose, forehead, mandible, palate, or eyes. Miscarriage was defined as spontaneous pregnancy loss in the first trimester or intrauterine fetal death (i.e., fetus loss after the third trimester). A timed follow-up was not applied for the mothers during pregnancy nor for the offspring after birth.

### 2.8. Risk of Bias and Quality Assessment

All included studies were assessed using QUADAS-2 (Quality Assessment of Diagnostic Accuracy Studies), a tool to evaluate the risk of bias and applicability of primary diagnostic accuracy studies. Any judgment regarding risk of bias is to be based on the following four domains: patient selection, index test, reference standard, and flow-timing. Judgment regarding applicability is based on the extent of which bias in any domain is likely to affect the question in the review. The risk of bias and applicability concerns were rated as “low”, “high”, or “unclear”. The percentage of papers with “high”, “low”, and “unclear” risk of bias and applicability concerns is graphically displayed in Figure 1. Adjunctively, according to AMSTAR 2 (A MeaSurement Tool to Assess systematic Reviews 2) score, “intermediate quality review” evaluation was obtained for this work [9,10].

### 2.9. Diagnostic Method of HPA

In almost all cases, both in the retrospective study and in the systematic review, maternal hyperphenylalaninemia (HPA) was identified by newborn screening, i.e., by tandem mass spectrometry on a blood spot taken from a heel prick. In a minority of cases, the disease was diagnosed differently, due to either the absence of newborn screening (pre-screening era) or the failure to diagnose by screening. In the latter case diagnosis was made by post-natal Phe level measurement in the suspect of HPA (because of suggestive clinical signs and/or positive family history).

### 2.10. Institutional Therapeutic Dietary Regimen

A rigorous low-phenylalanine diet is the mainstay of care for PKU patients. It could be the sole course of treatment or be combined with drugs. Dietary management has three objectives: limit the food amount of natural protein and phenylalanine to avoid the pathological rise of Phe in the blood and, consequently, in the brain; substituting natural proteins that have been cut out of the diet with synthetic proteins, amino acid mixtures/supplements, or protein substitute; reaching a normal growth stage [9,11] in order to keep blood phenylalanine concentrations within the target ranges specified by the European PKU Guidelines (from 120 to 360 μmol/L for women on a preconception diet or throughout pregnancy) [7]. PKU-affected women following dietary recommendations from preconception (pcD) or throughout pregnancy (pD), both in the retrospective study and in the systematic review, were restricted to consuming 25% or less of their daily intake of natural protein or phenylalanine. Phenylalanine tolerance was individually variable based on the severity of PKU (those with mild or moderate PKU tolerated more protein).

### 2.11. Statistical Analysis

Descriptive analyses were conducted, using frequencies and percentages for qualitative variables and median (IQR) for quantitative ones. As concerns the systematic review, Pearson’s chi-square test was used to calculate the *p*-value for independent variables. Therefore, a statistically significant difference was defined with a *p*-value < 0.05.

## 3. Results

### 3.1. Retrospective Study

This retrospective cohort study analyzed data from nine women diagnosed with phenylketonuria (PKU), focusing on maternal and neonatal outcomes based on different dietary interventions: “pcD” for diet started preconceptionally, “pD” for diet started during pregnancy, and “nD” for diet never started. Additionally, maternal phenylalanine (Phe) levels were categorized as “tP+” for Phe levels in target and “tP−” for non-target Phe levels. Here, we present the findings based on the collected data (Table 1).

### 3.2. Demographics and Diagnostic Method

The cohort comprised nine women with a mean age of 27.9 years at the time of their pregnancies. All women were diagnosed with PKU, with eight diagnosed through newborn screening and one through another unspecified method. This study reported a total of 14 pregnancies among the nine women. Out of the 14 pregnancies, 13 resulted in live births. There was one recorded miscarriage, suggesting a miscarriage rate of approximately 7.14% within the whole cohort.

### 3.3. Pregnancy Outcomes by Dietary Intervention

Women who started the PKU diet preconceptionally (pcD) had a total of 13 pregnancies. Out of these, 12 resulted in live births, and 1 in a miscarriage. There were no women who initiated the PKU diet during pregnancy (pD). Only one woman never started the PKU diet (nD), resulting in a live birth. Notably, the mean maternal Phe levels during pregnancies were recorded at 731.9 µmol/L (156 µmol/L for women tP+ on diet preconceptionally, 539.6 µmol/L for mothers tP− on diet before the beginning of the pregnancy, and 1500 µmol/L for women tP− who never started the diet).

### 3.4. Neonatal Outcomes by Maternal Phe Levels

Among pregnancies where maternal Phe levels were in target range (tP+), nine live births were recorded, with one case of miscarriage. In pregnancies where maternal Phe levels were not within the target range (tP−), there were four live births and no miscarriage. Only one case of IUGR/SGA was reported in pregnancies where maternal Phe levels were in the target range, whereas other adverse neonatal outcomes were not recorded; as we said above, all these women started the PKU diet preconceptionally. In pregnancies with non-target maternal Phe levels, adverse outcomes included two cases of congenital heart disease (CHD), including one transposition of the great vessels (TGV), one case of cleft lip and palate, three cases of IUGR/SGA, and one case of microcephaly.

### 3.5. Systematic Review

We identified 282 studies on PubMed: 110 by searching for “Maternal phenylketonuria”, 32 by “Maternal PKU syndrome”, and 140 by “Phenylketonuria and pregnancy”. Additionally, 29 studies were found by cross-references for a total of 311 papers. Then, we eliminated 30 duplicates and 45 non-English-language studies. Of the 236 remaining papers, 117 were excluded via title and abstract. Then, 119 full-text articles were evaluated for eligibility: 42 of them were eliminated after full-text reading. Finally, 77 studies [12,13,14,15,16,17,18,19,20,21,22,23,24,25,26,27,28,29,30,31,32,33,34,35,36,37,38,39,40,41,42,43,44,45,46,47,48,49,50,51,52,53,54,55,56,57,58,59,60,61,62,63,64,65,66,67,68,69,70,71,72,73,74,75,76,77,78,79,80,81,82,83,84,85,86,87,88] were included in our systematic review. Among them, ten manuscripts [18,27] are part of the Maternal Phenylketonuria Collaborative Study (MPKUCS) and reported data refer to the same cohort of PKU-affected women and their children. For this reason, these data were synoptically integrated and appear only once as total number of PKU-affected women, their diagnostic method, their type of hyperPhe, pregnancies, mean age at pregnancy onset, miscarriages, live births, malformations, and children with intellectual disability (Table 2). Search results following PRISMA methodology are shown in Figure 2.

### 3.6. Comprehensive Analysis of Dietary Management Timing and Maternal Phenylalanine Levels on Pregnancy and Neonatal Outcomes in PKU Patients

Phenylketonuria (PKU) poses significant challenges for women with regard to pregnancy management, requiring strict adherence to dietary protocols to optimize maternal and neonatal outcomes. This systematic review explores the intricate interplay between dietary management timing, maternal Phe levels, and their combined impact on pregnancy and neonatal outcomes in PKU patients. The dataset, encompassing 1068 women with PKU and 2094 pregnancies, categorizes once again the dietary interventions into preconceptional initiation (pcD), during pregnancy (pD), and never started (nD), further stratified based on maternal Phe levels in terms of target (tP+) or non-target (tP−).

### 3.7. Pregnancy Outcomes

Variations in miscarriage rates were observed across the distinct dietary management subcategories within the tP+ and tP− groups: the pcD subgroup within tP+ reported three miscarriages out of 378 pregnancies, indicating a low miscarriage rate. Within the pD subgroup, five miscarriages were documented among 253 pregnancies, reflecting a slightly higher rate compared to pcD. The nD subgroup within tP+ exhibited four miscarriages out of 76 pregnancies, indicating a higher miscarriage rate compared to pcD and pD subcategories.

The pcD subgroup within tP− showed three miscarriages out of 14 pregnancies, underscoring the impact of elevated Phe levels on pregnancy outcomes. In the pD subgroup within tP−, 25 miscarriages were recorded out of 321 pregnancies, highlighting the challenges faced by women with suboptimal Phe control. Within the nD subgroup of tP−, 321 miscarriages were reported out of 1000 pregnancies, emphasizing the detrimental impact of uninitiated dietary management on pregnancy outcomes.

### 3.8. Neonatal Outcomes

Neonatal outcomes, encompassing live births and the occurrence of congenital anomalies, displayed notable variations based on dietary management timing and maternal Phe levels.

### 3.9. Target Phe Levels (tP+) Group

The tP+ group had better overall outcomes, with fewer miscarriages and lower incidence of fetal anomalies.

Facial dysmorphisms were reported in 33 cases in the pcD group, 37 in the pD group, and 45 in the nD group. CHD incidence was the highest in the pD group (14 out of 249 born alive), followed by the nD group (1 out of 72), and pcD group (1 out of 376). IUGR/SGA incidence was slightly higher in the nD group. Microcephaly was most prevalent in the nD group (14 cases), followed by the pD (26 cases out of 249) and pcD (2 cases) groups. The rate of intellectual disability was recorded higher in the nD group compared to the other two groups. All these data are shown in Table 2.

### 3.10. Non-Target Phe Levels (tP−) Group

Higher rates of adverse outcomes, including miscarriages and congenital anomalies, were observed in the tP− group.

The pcD subgroup of tP− is small and this influences the significance of the data: 14 pregnancies resulted in 12 live births. The pD subgroup within tP− reported 297 live births out of 321 pregnancies. The nD subgroup had the lowest rate of live births, with 685 out of 1000 pregnancies resulting in live births, highlighting the challenges faced by women who never start diet before and throughout pregnancy and with suboptimal Phe control.

Among tP− newborns, facial dysmorphisms were reported in 5 cases in the pCD group, 192 in the pD group, and 122 in the nD group. CHD incidence was the highest in the pcD group (5 out of 12 born alive), followed by the pD group (50 out of 297), and nD group (46 out of 685). IUGR/SGA incidence was slightly higher in the pcD group, followed by nD and pD groups. Microcephaly was most prevalent in the pD group (126 cases out of 297), compared to the nD (255 cases out of 685) and the pcD (4 cases out of 12) groups. The rate of intellectual disability was recorded higher in the nD group than in the pD group, followed by the pcD group. All these data are shown in Table 2.

### 3.11. Statistical Analysis

Statistical analysis highlighted significant differences in pregnancy and neonatal outcomes based on the timing of dietary intervention and the management of maternal Phe levels. The *p*-values indicated strong statistical significance of the reduction in adverse outcomes in the tP+ group compared to the tP− one.

## 4. Discussion

The findings from this study shed light on the critical role of dietary management timing and maternal phenylalanine (Phe) levels in influencing pregnancy and neonatal outcomes among women with phenylketonuria (PKU). Before getting pregnant, women with PKU should follow a low-Phe diet to minimize risks associated with maternal PKU syndrome. Starting a low-Phe diet preconceptionally or early in pregnancy can prevent adverse outcomes such as congenital anomalies and developmental delays. Achieving satisfactory blood Phe control before or very early in pregnancy is linked to improved outcomes, including higher live birth rates and reduced neonatal sequelae [1,19,20,21,22,23,24].

Our data highlight that preconception dietary management (pcD) within target Phe levels (tP+) yields the most favorable pregnancy and neonatal outcomes, including lower miscarriage rates, fewer congenital anomalies, and better neurodevelopmental potential in offspring. Conversely, delayed or absent dietary management is associated with higher rates of miscarriages, congenital anomalies such as congenital heart defects (CHD) and microcephaly, and lower IQ scores in offspring. Importantly, even women who initiated dietary management during pregnancy (pD) experienced improved outcomes compared to those who never started (nD), though these benefits were less pronounced than with preconceptional intervention. This underscores the critical importance of proactive dietary planning and early intervention [26,86,88].

Mechanistically, high maternal Phe levels disrupt key metabolic and cellular pathways during fetal development, leading to neurodevelopmental and structural anomalies. Elevated Phe impairs the synthesis of neurotransmitters such as dopamine and serotonin, disrupts protein synthesis, and interferes with vascular development, particularly in the fetal brain and heart. These disruptions are most critical during early embryogenesis, a period when many women may not yet realize they are pregnant. Therefore, achieving and maintaining optimal Phe levels before conception is vital to prevent irreversible damage during these early stages of development [1,3].

Our retrospective study illustrates these findings through a detailed analysis of maternal Phe levels and outcomes across subgroups. In the pcD group within the tP+ category, neonatal outcomes were significantly better compared to those in other groups. Even within the tP− group, women in the pcD subgroup had relatively better outcomes than those in the nD group, suggesting that the timing of dietary management plays a crucial role independent of achieving target Phe levels. These results reinforce the importance of early and sustained dietary management in mitigating risks [1,19,20,21,22,23,24].

The systematic review further supports these conclusions, demonstrating that adherence to dietary protocols reduces the incidence of adverse outcomes, including microcephaly, congenital anomalies, and miscarriages. For example, the incidence of microcephaly was markedly lower in the tP+ subgroups, regardless of the specific dietary strategy employed, highlighting the importance of maintaining low maternal Phe levels throughout pregnancy. In contrast, the nD subgroup showed significantly higher rates of adverse outcomes across both target and non-target Phe level groups, reaffirming the severe repercussions of neglecting dietary management [26,86].

The relationship between maternal Phe levels and congenital anomalies such as CHD, dysmorphisms, and microcephaly highlights the necessity for ongoing education and proactive interventions in this patient population. Elevated Phe concentrations may exert teratogenic effects by disrupting fetal development through oxidative stress, metabolic imbalances, and alterations in cellular signaling pathways. These mechanisms underscore the biological rationale for stringent Phe control. Furthermore, evidence suggests that early neural development, including neural tube formation and synaptogenesis, is particularly vulnerable to high Phe levels, potentially leading to long-term neurodevelopmental deficits [26,86].

## 5. Conclusions

Our findings emphasize the critical interplay between the timing of dietary management, maternal Phe levels, and pregnancy outcomes in PKU patients. Proactive dietary management beginning preconceptionally and sustained throughout pregnancy significantly reduces the risk of miscarriage, congenital anomalies, growth restriction, and intellectual disability. Future research should explore the biological mechanisms underlying these associations, such as the role of Phe in disrupting neurotransmitter synthesis and vascular development, to further refine dietary strategies. Additionally, addressing practical barriers to adherence, such as logistical challenges, psychosocial factors, and gaps in patient education, is essential to improve compliance and outcomes. By enhancing preconceptional counseling and providing tailored support, we can optimize maternal and neonatal health in this vulnerable population [1,19,20,21,22,23,24,26,86].

### Study Limitations

The present study has several limitations: the small size of the sample composing the retrospective part; no data about the incidence of neuropsychiatric sequelae (such as intellectual disability) being reported in the retrospective study, since these elements have never been critically analyzed in our center. In addition, with regard to the systematic review, mean/median values of Phe during pregnancy cannot be given since these data are sometimes reported as mean, and sometimes as median in the studies under consideration. Additionally, it is possible to relate these data to the cut-offs imposed by current guidelines.

## Figures and Tables

**Figure 1 nutrients-17-00678-f001:**
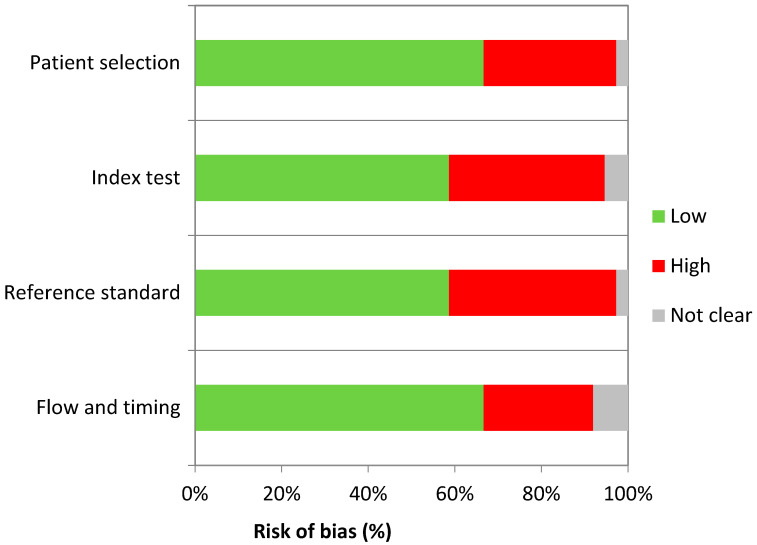
Percentage of studies with risk of bias and applicability concerns according to QUADAS-2 tool.

**Figure 2 nutrients-17-00678-f002:**
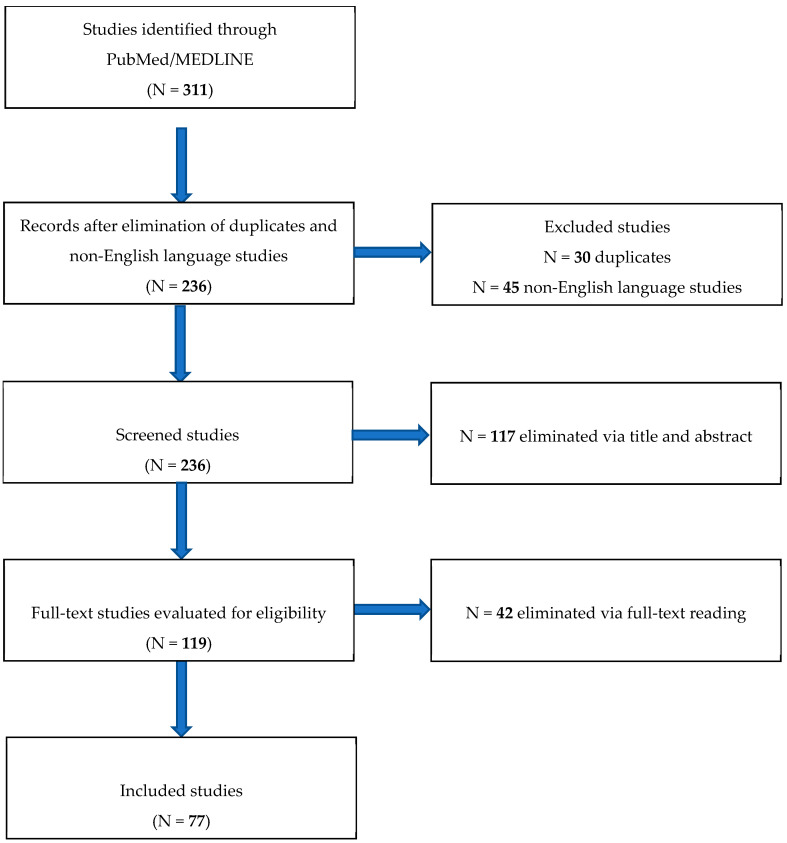
Search strategy following PRISMA methodology.

**Table 1 nutrients-17-00678-t001:** Retrospective studies on PKU patients performing or avoiding diet during pregnancy.

		tP−			tP+		
	Total (Retrospective Cohort)	nD	pD	pcD	nD	pD	pcD
N women	9						
N pregnancies	14	1	0	3	0	0	10
Age at pregnancy (years—mean)	27.9	22	n/a	30.9	n/a	n/a	30.8
Type of hyperPhe							
- classic PKU	12	1	n/a	2	n/a	n/a	9
- other than classic PKU	2	0	n/a	1	n/a	n/a	1
Mean Phe levels during pregnancy (μmol/L)	731.9	1500	n/a	539.6	n/a	n/a	156
N miscarriages	1	0	n/a	0	n/a	n/a	1
N born alive	13	1 *	n/a	3	n/a	n/a	9
Dysmorphisms	1	1 *	n/a	0	n/a	n/a	0
CHD	2	1 *	n/a	1 **	n/a	n/a	0
IUGR–SGA	4	1 *	n/a	2	n/a	n/a	1
Microcephaly	1	0	n/a	1	n/a	n/a	0

* dead at 4 months of age for MPKUS (clef lip and palate and CHD); ** transposition of the great vessels (TGV). Legend: CHD: congenital heart disease; IUGR: intrauterine growth restriction; n/a: not available; nD: patients not starting diet in pregnancy; pcD: patients starting diet from preconception; pD: patients starting diet from pregnancy; Phe: phenylalanine; PKU: phenylketonuria; SGA: small for gestational age; tP−: patients who did not meet the target phenylalanine values during pregnancy (120–360 μmol/L); tP+: patients who showed target phenylalanine values during pregnancy.

**Table 2 nutrients-17-00678-t002:** Systematic reviews on PKU patients avowing diet in pregnancy.

			tP−			tP+			Unknown	
		Total (Systematic Review)	nD	pD	pcD	nD	pD	pcD	Unknown	*p*-Value
	N women	1068								
	N pregnancies	2094 *	1000 **	321 ^§^	14 ^#^	76	253 ***	378 ^#^	52	n/a
	Age at pregnancy	n/a	n/a	n/a	n/a	n/a	n/a	n/a	n/a	n/a
Type of maternal HPA	Classic PKU	917	225	227	12 °	0	166	257	30	n/a
	Other than classic PKU	425	90	74	1	75	76	109	0	n/a
	Unknown	752	685	20	1	1	11	12	22	n/a
Type of delivery	Miscarriages	375	321	25	3	4	5	3	14	<0.05
	Born alive	1729	685	297	12	72	249	376	38	<0.05
Facial dysmorphisms	yes	434	122	192	5	45	37	33	0	<0.05
	unknown	406	318	19	1	26	10	2	30	n/a
CHD	yes	117	46	50	5	1	14	1	0	<0.05
	unknown	30	0	0	0	0	0	0	30	n/a
IUGR–SGA	yes	181	131	17	3	2	6	3	19	<0.05
	unknown	98	20	15	0	0	0	33	30	n/a
Microcephaly	yes	441	255	126	4	14	40	2	0	<0.05
	unknown	30	0	0	0	0	0	0	30	n/a
Intellectual disability	yes	472	311	121	4	16	12	1	7	<0.05
	unknown	432	172	103	2	40	29	48	38	n/a

* 4 sets of twins and 6 twin pregnancies; ** 2 sets of twins; *** 1 twin pregnancy; ^#^ 1 twin pregnancy, 1 stillbirth occurred because of intrapartum cord entanglement, and 1 neonate died of sudden infant death syndrome at the age of 4 weeks; ^§^ 1 twin pregnancy; ° 1 twin pregnancy. Legend: CHD: congenital heart disease; IUGR: intrauterine growth restriction; n/a: not available; nD: patients not starting diet in pregnancy; pcD: patients starting diet from preconception; pD: patients starting diet from pregnancy; PKU: phenylketonuria; SGA: small for gestational age; tP−: patients who did not met the target phenylalanine values during pregnancy (120–360 μmol/L); tP+: patients who showed target phenylalanine values during pregnancy.

## Data Availability

The data supporting the reported results can be found at the University Hospital “Policlinico-San Marco”, Catania, Italy.

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
