# Peer review of "Maternal Phenylketonuria and Offspring Outcome: A Retrospective Study with a Systematic Review of the Literature"

_nutrients, 2025, doi:10.3390/nu17040678_

Round 1
Reviewer 1 Report
Comments and Suggestions for Authors
The authors present a study on PKU in pregnancy and effects of maternal PKU syndrome (MPKUS) on the offspring. The study consists of two parts: A retrospective evaluation of a cohort of own patients and a systematic review of literature.
While this is an important subject the significance of this manuscript remains quite limited. One reason is that the own cohort is very small (9 patients and 16 pregnancies) and the data are incomplete. In two cases the pregnancy outcome is not mentioned at all (13 live births, 1 miscarriage, no explanation why this does not add up to 16). It is also not clear how many patients were affected by complications typical for MPKUS as only the the number of symptoms is given.
Secondly, the systematic literature review is hampered by the fact that only two queries ("Maternal phenylketonuria" and "Maternal PKU syndrome") were used. Most of the references identified by this strategy are older than 20 years, the most recent publication dates from 2015 (!). By using the query „Phenylketonuria AND pregnancy“, for example, 937 publications including more recent studies such as PMID 36054426 or 36843352 would have been identified.
The focus of the authors is on dietary measures only to normalize Phe concentrations, drug therapy with sapropterin or pegvaliase is not taken into account and just briefly mentioned in the introduction. As they play an increasing role in PKU pregnancies, that could have been an additional focus to improve novelty of the manuscript. In this case, however, references published later than 2015 would have to be included.
Introduction: The introduction could be shortened by omiting information that not relevant for this study, such as PKU incidence (not „prevalence“) in different regions, historical differentiation of PKU phenotypes (lines 47-51).
Material and methods: The study focuses on two groups of patients with „Phe values in target range“ and „Phe values not in target range“. However, it remains unclear, which values were considered for this classification: All values throughout pregnancy, more than a certain percentage of values or median/mean values within target range? If the latter was chosen then patients in the tP+ cohort still may have had elevated concentrations in the vulnerable phase of organ formation.
Results: The authors calculate p-values and report „statistical significance“ in cohorts with very small sample sizes lacking statistical power (for examle: Table 1: miscarriages n=1, p=1). This is misleading and not scientifically sound. The tables are unclear and confusing because the cohorts are characterized by abbreviations only and are not clearly seperated from each other. The information if PKU was identified in neonatal screening or differently should not be part of the table. Table 2 is slightly better structured because an additional first column is used for some basic classifications. The legend of Table 2 is far too long, the information should be given in the text. The „Results“ text should be considerably shortened and be limited to relevant information. The 12 (!) subheadings in this part should be summarized to fewer seperate parts. Redundancies (i.e. information in line 328-332 already been given in lines 304-309) should be omitted as well as and passages that belong into introduction (lines 294 – 297) or discussion (lines 312 or 330).
Discussion: The discussion is poor and revolves around a single statement „preconceptional start of therapy is necessary to ensure optimal outcome for the child“, which is not new. New aspects or limitations of the study are not mentioned.
Comments on the Quality of English LanguageThe wording is cumbersome and not easy for readers (i.e. line 123 "start of diet from pregnancy" instead of "diet started in pregnancy"; line 114 "a massive clinical ... evaluation of the newborn was made" instead of "the newborn was evaluated clinically and auxologically").
There are several typos: e.g. lines 453 and 473: "Congenita" instead of "congenital" and "Intra-Uterin" instead of "intrauterine"
Author Response
Reviewer 1
The authors present a study on PKU in pregnancy and effects of maternal PKU syndrome (MPKUS) on the offspring. The study consists of two parts: A retrospective evaluation of a cohort of own patients and a systematic review of literature.
While this is an important subject the significance of this manuscript remains quite limited. One reason is that the own cohort is very small (9 patients and 16 pregnancies) and the data are incomplete. In two cases the pregnancy outcome is not mentioned at all (13 live births, 1 miscarriage, no explanation why this does not add up to 16). It is also not clear how many patients were affected by complications typical for MPKUS as only the the number of symptoms is given.
Secondly, the systematic literature review is hampered by the fact that only two queries ("Maternal phenylketonuria" and "Maternal PKU syndrome") were used. Most of the references identified by this strategy are older than 20 years, the most recent publication dates from 2015 (!). By using the query „Phenylketonuria AND pregnancy“, for example, 937 publications including more recent studies such as PMID 36054426 or 36843352 would have been identified.
Authors’ reply: Thank you for the concerns you raised. I will respond to your criticism point by point. It is true that the sample of our retrospective study is small, but we believe that it adds value to the literature, also in anticipation of any future reviews on the topic.
The data on our cohort are not incomplete. We do not reach number 16 because 2 pregnancies were still in progress at the time of writing the manuscript. Regarding the data we reported in the retrospective study, we provide, similar to the systematic review, only the number of children with a certain manifestation of MPKUS (e.g., one had microcephaly). We have added on your instruction a clear reference to the exclusion of work reporting therapies other than dietary, such as sapropepterin or pegvaliase. We thought that more stringent word queries would narrow the search field and avoid selecting articles that did not respond to our search. In any case, this is the largest systematic review to date on the topic of MPKUS.
The focus of the authors is on dietary measures only to normalize Phe concentrations, drug therapy with sapropterin or pegvaliase is not taken into account and just briefly mentioned in the introduction. As they play an increasing role in PKU pregnancies, that could have been an additional focus to improve novelty of the manuscript. In this case, however, references published later than 2015 would have to be included.
Authors’ reply: Our study focuses on the effects of diet on the incidence of MPKUS and its various manifestations. So, we only report data from pregnant women on diet and who were unable or unwilling to access other treatment options, such as sapropeterin or pegvaliase.
Introduction: The introduction could be shortened by omiting information that not relevant for this study, such as PKU incidence (not „prevalence“) in different regions, historical differentiation of PKU phenotypes (lines 47-51).
Authors’ reply: Thank you for your suggestion. The introduction has been shortened: we believe that the introduction will provide interesting notes for readers.
Material and methods: The study focuses on two groups of patients with „Phe values in target range“ and „Phe values not in target range“. However, it remains unclear, which values were considered for this classification: All values throughout pregnancy, more than a certain percentage of values or median/mean values within target range? If the latter was chosen then patients in the tP+ cohort still may have had elevated concentrations in the vulnerable phase of organ formation.
Authors’ reply: At the beginning of “Patient Group of the retrospective analysis” section we now stated that patients are considered in target when Phe values are between 120-360 micromol/L, as explicitly stated in the guidelines. Values above this cut-off are considered not in target, as defined in our materials and methods section.
Results: The authors calculate p-values and report „statistical significance“ in cohorts with very small sample sizes lacking statistical power (for examle: Table 1: miscarriages n=1, p=1). This is misleading and not scientifically sound. The tables are unclear and confusing because the cohorts are characterized by abbreviations only and are not clearly seperated from each other. The information if PKU was identified in neonatal screening or differently should not be part of the table. Table 2 is slightly better structured because an additional first column is used for some basic classifications. The legend of Table 2 is far too long, the information should be given in the text. The „Results“ text should be considerably shortened and be limited to relevant information. The 12 (!) subheadings in this part should be summarized to fewer seperate parts. Redundancies (i.e. information in line 328-332 already been given in lines 304-309) should be omitted as well as and passages that belong into introduction (lines 294 – 297) or discussion (lines 312 or 330).
Authors’ reply: We’ve fixed the tables in accordance with your suggestions, removing the useless information (screening method). We have reduced the footnotes to make them more usable and less redundant. We have reduced subheadings in table 2, with only six remaining as they make the table more useful: their permanence does not detract from the quality of the work and the data reported. We have removed conclusions from the results section, in order to make it lighter and removed the redundancies in the same results section.
Discussion: The discussion is poor and revolves around a single statement „preconceptional start of therapy is necessary to ensure optimal outcome for the child“, which is not new. New aspects or limitations of the study are not mentioned.
Authors’ reply: The discussion has been totally rephrased and now include a new paragraph with the limitations, as suggested be reviewer 1.
The wording is cumbersome and not easy for readers (i.e. line 123 "start of diet from pregnancy" instead of "diet started in pregnancy"; line 114 "a massive clinical ... evaluation of the newborn was made" instead of "the newborn was evaluated clinically and auxologically").
Authors’ reply: We’ve edited the text accordingly.
There are several typos: e.g. lines 453 and 473: "Congenita" instead of "congenital" and "Intra-Uterin" instead of "intrauterine"
Authors’ reply: Typos were fixed where present.
Reviewer 2 Report
Comments and Suggestions for Authors
Dear Authors
Thank you for your contribution and work it has a very clear and important message
There are a few areas that I have added to the PdF which you will see I have questioned
In some parts the English is a little 'stiff' and too formal and there are a few sentences that need to be re worded. But overall it reads very well and to write this as a second language has to be applauded
I think the tables should go nearer to the text to make it easier to understand
The discussion is a little repetitive in part and could be condensed giving more room to discuss the reasons why high Phe leads to congenital abnormalities, giving a more scientific input to your work and reasons for lower Phe concentrations pre -conception.

I have added on the PdF my comments regarding the English and where small adjustments are needed
Author Response
Reviewer 2
Dear Authors
Thank you for your contribution and work it has a very clear and important message
There are a few areas that I have added to the PdF which you will see I have questioned
In some parts the English is a little 'stiff' and too formal and there are a few sentences that need to be re worded. But overall it reads very well and to write this as a second language has to be applauded
Authors’ reply: Thank you very much for your appreciation. We have modified many typos and rephrased the text: we hope that now the text can be easily usable for readers and that it reports the most relevant information on MPKUS in pregnant women not on a diet or only on a diet, excluding those on drug therapy.
I think the tables should go nearer to the text to make it easier to understand
Authors’ reply: Table are placed after the text following the suggestion of academic reviewer.
The discussion is a little repetitive in part and could be condensed giving more room to discuss the reasons why high Phe leads to congenital abnormalities, giving a more scientific input to your work and reasons for lower Phe concentrations pre -conception.
Authors’ reply: Discussion has been condensed and deeply rephrased, avoiding some repetitions (from the systematic review to the retrospective analysis). A new subchapter on study limitations has been added.
Reviewer 3 Report
Comments and Suggestions for Authors
Congratuklations.
Two minimal consdierations:
1. In the introduction to add a sentences clarifying (for those not familiar to the disease) that the reason for the diet in the mother is independently of the statuts on PKU of the fetus.
2. lines 112: this is the first time the abbreviations appear (they are explained in the next paragraph) Plesae change,
Author Response
Reviewer 3
Congratulations.
Two minimal consdierations:
- In the introduction to add a sentences clarifying (for those not familiar to the disease) that the reason for the diet in the mother is independently of the statuts on PKU of the fetus.
Authors’ reply: Thank you for the suggestion: we have added this sentence in the third paragraph of the introduction.
- lines 112: this is the first time the abbreviations appear (they are explained in the next paragraph) Plesae change,
Authors’ reply It has been changed according to your note.
Round 2
Reviewer 1 Report
Comments and Suggestions for Authors
The revised manuscript is an improvement on the previous version but there are still points of concern that need to be adressed. Also, several aspects mentioned in my previous review have not been processed adequately.
1. Patients who have not given birth yet have to be excluded from a study focused on neonatal outcomes as there is no data on neonatal outcomes.
2. The explanation of the authors, why only a very narrow search strategy has been used, is not convincing. The two recent publications on PKU and pregnancy I recommended to include (PMID 36054426 and 3684335) did only include pregnant women on diet or without therapy. To me it seems as if the authors wanted to avoid the work involved in a new literature search.
3. The authors did not answer my question if the classification „Phe values in target range“ means that in the retrospective study all values throughout pregnancy, more than a certain percentage of values or median/mean values were within the range 120-360 µmol/l. This has to be clarified and mentioned in „Patient groups of the retrospective study“.
4. Line 116, 117: The recommendation mentioned in the cited references refers to pregnant women, not to „fertile women“. This should be corrected.
5. Table 1 is much clearer now. However, the column containing p-values has to be omitted as statistical power is lacking (as stated in my previous review). Consequentially, results of the retrospective study can only be evaluated descriptively, the paragraph „Statistical Analysis“ should be omitted.
6. Line 419: Please correct typo („Phe“ instead of „The“)
7. The last sentence (line 421-422) should be rephrased. If Phe levels are not mentioned in publications and only information such as „high“ or „low“ are given, they cannot be related to cutoffs in current recommendations who were developed much later.
Author Response
To Prof. Maria Luz Fernandez,
Department of Nutritional Sciences,
University of Connecticut, Storrs, USA
Editor-in-chief
Nutrients
Dear Editor,
We sincerely appreciate the opportunity to revise and resubmit again our manuscript entitled “Maternal phenylketonuria and offspring outcome: a retrospective study with systematic review of the literature.” to Nutrients. We are grateful to the reviewer for his thoughtful and constructive comments, which have greatly helped us improve the quality and clarity of our work.
In this revised version, we have carefully addressed all the points raised by the reviewers and have incorporated their suggestions into the manuscript (highlighted in light blue). Below, we provide a detailed, point-by-point response to each comment, specifying the changes made and referencing the relevant sections of the revised manuscript.
Reviewer 1
The revised manuscript is an improvement on the previous version but there are still points of concern that need to be adressed. Also, several aspects mentioned in my previous review have not been processed adequately.
- Patients who have not given birth yet have to be excluded from a study focused on neonatal outcomes as there is no data on neonatal outcomes.
Authors’ reply: Patients who have not given birth have been excluded.
- The explanation of the authors, why only a very narrow search strategy has been used, is not convincing. The two recent publications on PKU and pregnancy I recommended to include (PMID 36054426 and 3684335) did only include pregnant women on diet or without therapy. To me it seems as if the authors wanted to avoid the work involved in a new literature search.
Authors’ reply: Thank you very much for this suggestion. A new literature research has been performed, according to the original filters of our systematic review. The two articles you recommended to include are now included in the revision.
- The authors did not answer my question if the classification „Phe values in target range“ means that in the retrospective study all values throughout pregnancy, more than a certain percentage of values or median/mean values were within the range 120-360 µmol/l. This has to be clarified and mentioned in „Patient groups of the retrospective study“.
Authors’ reply: This point has been clarified in the section re the retrospective study.
- Line 116, 117: The recommendation mentioned in the cited references refers to pregnant women, not to „fertile women“. This should be corrected.
Authors’ reply: We have performed this corrections in the text.
- Table 1 is much clearer now. However, the column containing p-values has to be omitted as statistical power is lacking (as stated in my previous review). Consequentially, results of the retrospective study can only be evaluated descriptively, the paragraph „Statistical Analysis“ should be omitted.
Authors’ reply: The column of the table and the paragraph have been omitted.
- Line 419: Please correct typo („Phe“ instead of „The“)
Authors’ reply: This typo has been corrected.
- The last sentence (line 421-422) should be rephrased. If Phe levels are not mentioned in publications and only information such as „high“ or „low“ are given, they cannot be related to cutoffs in current recommendations who were developed much later.
Authors’ reply: The last sentence has been rephrased accordingly.
We trust that the revisions have adequately addressed the concerns raised and that the manuscript is now suitable for publication in Nutrients.
Thank you for your consideration, and please do not hesitate to contact us if further clarifications are needed.
Prof. Andrea D. Praticò
University Kore of Enna
Enna, Italy